# Treatment of Acute and Long-COVID, Diabetes, Myocardial Infarction, and Alzheimer’s Disease: The Potential Role of a Novel Nano-Compound—The Transdermal Glutathione–Cyclodextrin Complex

**DOI:** 10.3390/antiox13091106

**Published:** 2024-09-12

**Authors:** Ray Yutani, Vishwanath Venketaraman, Nisar Sheren

**Affiliations:** 1Department of Family Medicine, College of Osteopathic Medicine of the Pacific, Western University of Health Sciences, Pomona, CA 91766, USA; 2Department of Basic Medical Sciences, College of Osteopathic Medicine of the Pacific, Western University of Health Sciences, Pomona, CA 91766, USA; vvenketaraman@westernu.edu; 3College of Osteopathic Medicine of the Pacific, Western University of Health Sciences, Pomona, CA 91766, USA; nisar.sheren@westernu.edu

**Keywords:** oxidative stress, reactive oxygen species, antioxidants, glutathione, glutathione–cyclodextrin complex

## Abstract

Oxidative stress (OS) occurs from excessive reactive oxygen species or a deficiency of antioxidants—primarily endogenous glutathione (GSH). There are many illnesses, from acute and post-COVID-19, diabetes, myocardial infarction to Alzheimer’s disease, that are associated with OS. These dissimilar illnesses are, in order, viral infections, metabolic disorders, ischemic events, and neurodegenerative disorders. Evidence is presented that in many illnesses, (1) OS is an early initiator and significant promotor of their progressive pathophysiologic processes, (2) early reduction of OS may prevent later serious and irreversible complications, (3) GSH deficiency is associated with OS, (4) GSH can likely reduce OS and restore adaptive physiology, (5) effective administration of GSH can be accomplished with a novel nano-product, the GSH/cyclodextrin (GC) complex. OS is an overlooked pathological process of many illnesses. Significantly, with the GSH/cyclodextrin (GC) complex, therapeutic administration of GSH is now available to reduce OS. Finally, rigorous prospective studies are needed to confirm the efficacy of this therapeutic approach.

## 1. Introduction

As our primary intracellular antioxidant, GSH is integral in maintaining our health and well-being against oxidative stress (OS) [1]. OS results from a high level of reactive oxygen species (ROS) relative to antioxidants. Although ROS is beneficial at physiologic levels, its excess leads to OS. OS plays a role in our immune response (innate, adaptive, and autoimmune) that may be physiologic or pathologic [2,3,4]. OS and GSH deficiency are associated with advanced age and diseases of the brain, heart, lungs, eyes, liver, kidneys, and of the nervous and vascular systems [5,6,7,8,9,10]. OS-related injuries can be acute (infections, systemic inflammatory response syndrome, and ischemia-reperfusion injury) or chronic (idiopathic pulmonary fibrosis, atherosclerosis, diabetes mellitus, and neurodegenerative diseases) [1,5,9,11,12,13,14]. Physiologically, OS is suppressed using antioxidants, primarily endogenous GSH, our body’s “master antioxidant” [1,9,15]. OS may not be a marker but instead an initiator or promoter of these diseases. Thus, inhibiting the development of OS would be simpler and more effective than addressing each specific product of the subsequent complex and dynamic inflammatory response. Finally, providing therapeutic levels of exogenous GSH could address the pathophysiologic processes of these diseases and insults before severe or irreversible injury occurs [8,11,16]. Evidence is presented that a novel nano-agent, the GSH/cyclodextrin (GC) complex, can effectively increase GSH levels and counter OS [17]. With this complex, GSH is incorporated in gamma-cyclodextrin (γ-CD), is easily administered topically, and avoids the limitations of oral, IV GSH, and N-acetyl cysteine (NAC) [5,11,15,16,18]. In contrast to the G/C complex, NAC does not increase GSH levels unless there is a depletion of GSH and the presence of a functional enzymatic pathway for de novo synthesis of GSH [19]. Hopefully, delivering GSH directly as the complex will prompt studies to determine the efficacy of exogenous GSH in suppressing OS. Significantly, GHS deficiency may not only be related to aging and diseases, but restoring GSH could be a potential therapeutic target for many illnesses [20].

## 2. Oxidative Stress (OS)

### 2.1. Reactive Oxygen Species and OS

At a physiologic level, ROS regulates aging and activates the innate inflammatory response [2]. This innate response, in turn, augments the secondary adaptive immune response, as shown in Figure 1 [4]. ROS and reactive nitrogen species (RNS) include both radical forms (e.g., superoxide, hydroxyl radical, and nitric oxide) and non-radical forms (e.g., hydrogen peroxide, hypochlorous acid, and peroxynitrite) [4,14]. Although the non-radical species are not directly damaging to cells, they cause oxidative damage when converted to their radical form (usually after reacting with metal ions). As signaling molecules, ROS/RNS regulate or mediate many physiologic functions. These include gene activation, cellular growth, and blood pressure control [14]. The duality of ROS in diseases, whether physiologically beneficial or pathologically harmful, has been outlined [7]. The “Goldilocks” principle of adaptive vs. deleterious ROS response using an exercise model has been presented [21]. The level of ROS, which in excess leads to OS, appears to be the determining factor.

OS occurs when the biological system creates excessive ROS and, to a lesser extent, RNS, which alters the normal homeostatic balance between pro-oxidant and antioxidant molecules, as shown in Figure 2. ROS/RNS are produced in response to external stimuli. OS damage to DNA and other biomolecules may impair normal functions of tissue cells and lead to human aging and disease [1,9]. Under adaptive physiologic conditions, excessive ROS can be controlled by enzymatic (superoxide dismutase, catalase, glutathione peroxidase) and non-enzymatic (glutathione) antioxidants. GSH is well-established as the most important cellular antioxidant [1,9,11,15,22]. ROS is primarily generated in mitochondria, but it is also produced in peroxisomes, endoplasmic reticulum, and lysosomes [14].

Adding to the complexity of managing OS, the influence of biochemical enzymes should be noted, specifically glutathione S-transferases (GSTs), glutathione peroxidases (GPx), and peroxiredoxin (Prx) [23]. GSTs’ major function is in cellular protection by catalyzing the conjugation of GSH with many hydrophobic and electrophyllic intermediates [24]. These intermediates include many carcinogens, therapeutic agents, and products of OS. The conjugation process depletes the available endogenous GHS, as manifested in acetaminophen toxicity and likely COVID-19 illness.

GPx is a family of enzymes (GPx1–GPx8) to combat OS. Each has a unique mechanism of action and site of action [25]. The GPx family plays a role in combating OS, tumor development, viral infection, obesity, apoptosis, signal transduction, and inflammation. GPx4 is a critical enzyme for destroying fatty acid hydroperoxidases and mediates the process of ferrozois (cell death from lipid peroxidation). GPx3 has been associated with OS during renal ischemia-reperfusion injuries [26]. Through an in vivo murine model, the overexpression of GPx1 and the development of insulin resistance and obesity were associated [27].

Peroxyredoxins (Prx) are a family of peroxidase enzymes that have a significant role in regulating peroxide levels within cells. There is evidence that they reduce 90% of cellular peroxides [28]. Peroxyredoxin 2 (Prx2), together with catalase and GPx, through their binding to the red blood cell membrane, has been associated with hemolytic anemias (in non-immune hemolytic anemia, hereditary spherocytosis, sickle cell disease, β-thalassemia, and healthy individuals) [29].

Finally, gluththionylation of proteins (a redox-mediated posttranslational modification that regulates the function of target proteins by conjugating GSH with a cysteine thiol group on the target protein) has a protective effect of GSH against OS in many diseases has been presented [30]. Its role in the pathogenesis of many neurodegenerative diseases was reviewed [31]. It remains to be determined if therapeutic administration of GSH can influence these many enzymatic processes.

There are many human disorders clinically linked to OS [1,5,9,10,14,32]. These include those of the brain (e.g., Alzheimer’s, Parkinson’s, amyotrophic lateral sclerosis, multiple sclerosis, autism, encephalitis, and both thrombotic and hemorrhagic stroke), the liver (nonalcoholic steatohepatitis—NASH and hepatic encephalopathy), the pancreas (pancreatitis, diabetes, and its complications), the lungs (COVID-associated ARDS, idiopathic pulmonary fibrosis, cystic fibrosis), the eyes (macular degeneration, retinitis pigmentosa), pathologic aging (Huntington-Gilford progeria syndrome, Werners syndrome), certain cancers (skin, kidney, bowel, and breast), and the skin (melanin abnormalities). Additional disorders include organ failure (renal, hepatic), age-related frailty (“inflammaging”), and disorders of the cardiovascular system (hypertension, dyslipidemia, atherosclerosis, myocardial infarction, restenosis, myocarditis, ischemia/reperfusion injury, and thrombosis), autoimmunity (rheumatoid arthritis, fibromyalgia), and the inhibition of viral replication (HIV, influenza and herpes simplex) [1,3,5,8,9,10,11,12,33,34,35]. Examples of the influence of OS on diseases are illustrated in Figure 3.

There is an inverse relationship between GSH level and OS in diseases [9,15,16]. Congenital GSH synthetase deficiency leads to severe consequences (recurrent bacterial infections, mental retardation, motor disturbances, retinal pigmentation) that highlights the pathologic sequelae of GSH deficiency. The brain is highly susceptible to OS, likely due to high oxygen consumption and a low GSH level compared with other tissues. However, the limitations of current therapies have been highlighted [9].

### 2.2. Immunothrombosis and Thromboinflammation and OS

A significant response to OS is immunothrombosis, and its inflammatory consequence is known as thromboinflammation. Although the activation of the coagulation cascade is our natural hemostatic defense mechanism, pathologic immune activation can lead to numerous thrombotic diseases of the macrovascular and microvascular systems [36]. Diseases associated with pathologic thrombosis include those of the veins, coronary arteries, and lungs [16,33,36,37,38,39,40,41]. Immunothrombosis is activated as a defense response to bacterial or viral infection and other immune response triggers [36,41]. Macrovascular events include myocardial infarction, stroke, venous thromboembolism, and pulmonary embolism. Microvascular thrombosis is involved in sepsis, ischemia-reperfusion injury (IRI), organ transplantation rejection, major trauma, severe burns, antiphospholipid syndrome, preeclampsia, and sickle cell disease [36,40]. 

The consequent thromboinflammation causes the activation of platelets and innate immune cells that initiates a coagulation cascade involving the complement system (C’) and tissue factor (TF) [16,33,39,40]. This process is a prominent feature of ARDS from severe COVID-19 [16]. Severe respiratory symptoms of COVID-19 illness are associated with elevated D-dimer and von Willebrand Factor (vWF) levels, complement activation, and antiphospholipid antibodies [16,33]. Additionally, in arterial thrombosis, OS-induced endothelial injury initiates a coagulation cascade (involving tissue factor, collagen, and (vWF) that leads to thrombin formation [33,42].

Thromboinflammation occurs in both acute and long-COVID diseases of the liver, lungs, and the cardiovascular system [33,43,44,45,46,47,48,49,50]. The developmental sequence has been presented [43,44,51]. It is triggered by external insults (e.g., tobacco, asbestos/silica, radiation, viruses, or bleomycin) followed by macrophage activation, CS, pneumocyte apoptosis, fibroblast recruitment, and collagen deposition. OS is a significant molecular component of fibrosis [45,51,52]. Idiopathic pulmonary fibrosis (IPF), a rare disease without a known initiator, has been noted to have similarities with lung fibrosis from severe COVID-19 and post-COVID pulmonary fibrosis (PCPF) [43,44,52].

The risk of venous thromboembolism increases with age and is influenced by OS in red blood cells (RBCs). High levels of ROS disturb RBC membrane structure and function, resulting in loss of membrane integrity and decreased deformity that produces a hypercoagulable state marked by enhanced RBC aggregation, RBC binding to endothelial cells (causing impairment of nitric oxide (NO)-dependent vasodilatation), RBC-induced activation of platelets, complement, and TF [12,16,39,41]. 

In addition to the process of atherosclerosis, ROS is involved in other pathologic cardiovascular events, including neointima formation after balloon angioplasty with stenting and possibly the development of abdominal aortic aneurysm [53]. The International Society on Thrombosis and Haemostasis Congress presented an overview of the distinction between immunothrombosis and thromboinflammation [54]. Although both involve platelet aggregation, the former primarily involves the venous system (e.g., cerebral venous thrombosis) and the latter the arterial system (e.g., ischemic stroke). However, it appears that they are clinical “bookends” of an integrated pathophysiologic continuum.

### 2.3. The Complement System and OS

The integral role of the complement system in macro- and micro-thrombosis have been thoroughly reviewed and illustrated [36,40,41]. The role of OS to initiate the complement response has been firmly established in diseases of the retinal pigment epithelium [55]. 

## 3. OS-Associated Diseases

### 3.1. COVID-19

As previously mentioned, GSH plays a significant role in the immunothrombotic process involving multiple organs and the vascular system, particularly COVID-19 [9,15,16,56]. GSH deficiency and OS are involved in the pathologic consequences of acute and chronic COVID-19 illness [15,16,57,58]. Acute COVID induces an unchecked cytokine cascade leading to the cytokine storm (CS) that is driven by OS [59]. Although not firmly established, post-COVID complications apparently involve a persistent antigenic stimulus leading to what is commonly known as long-COVID and officially as post-acute sequelae of COVID-19 (PASC), arterial thrombosis, and microvascular thrombosis associated with activation of vWF [16,59,60]. vWF is not only a biomarker of severe acute pulmonary injury but also of persistent respiratory symptoms in long-COVID [61]. Finally, complement dysregulation has been associated with long-COVID 57]. 

Using COVID-19 as a model, the significance of GSH as a critical antioxidant against OS, the association of its depletion by oxidation, and its decreased synthesis have been reviewed [15,16]. This decrease is inversely related to the severity of acute and chronic illnesses. GSH deficiency is associated with TGFβ activity in severe COVID-19 illness. TGFβ causes low levels of GSH by repressing the enzyme γ-glutamyl-cysteine ligase (GCL) required for the de novo synthesis of GSH [15,16,62]. Impaired synthesis is also driven by hypoxia, which decreases the two enzymes that catalyze the de novo synthesis of GSH (GCL and GS) [15]. Significantly, the lack of studies using GSH was recognized. [63] This exposes the current inability to deliver GSH safely, efficiently, and therapeutically clinically. The efficacy of a novel topical GSH delivery system that appears to overcome this problem has been published [17]. Through nanotechnology, GSH has been incorporated into cyclodextrin (CD) to produce the GSH/CD (GC) complex. This preliminary study with the complex demonstrated its efficacy in young, healthy individuals to (1) deliver GSH into RBCs, (2) effect OS as demonstrated to lower lipid peroxidation as measured using malondialdyhyde (MDA), and (3) inhibit *Mycobacterium avium* in vitro. However, its use in large, rigorous clinical trials is needed to confirm if GSH, using the G/C complex, can address OS in current untreatable or sub-optimally treated illnesses. Significantly, it has been confirmed that cyclodextrin can cross the blood–brain barrier (BBB) [64]. As previously stated, the place of the complex in disease management would be its early use in acute illness to address OS before it leads to irreversible tissue damage and its continued use in chronic illness.

### 3.2. Acute COVID-19

The complex nature of the immuno-thrombotic process has been detailed in previous reviews [15,16,54,65]. Additionally, the inverse relationship of OS with a low glutathione level has been well acknowledged [15,16,58,66,67]. A low GSH level is associated with corresponding COVID-19 disease severity in elderly individuals and those with diabetes, hypertension, and obesity [1,15,56,62,68]. SARS-CoV-2 virus stimulates ROS production through multiple pathways, leading to enhanced production of pro-inflammatory cytokines and the subsequent cytokine storm (CS) [69,70]. Pro-inflammatory cytokines stimulate the inflammatory process by activating the nuclear kappa B (NF-κB) signaling pathway, resulting in a cyclical positive feedback loop [69]. 

The severe symptoms of acute COVID-19 illness, particularly ARDS, are a consequence of thromboinflammation resulting from immune dysregulation, inflammation, and endothelial cell damage [15,16]. The dysregulated immune response leads to a hyperactive innate immune and hypercoagulable state that involves endothelial damage followed by platelet aggregation and a coagulation cascade involving fibrin and thrombin that is initiated by activated von Willebrand Factor (vWF). A fibrin degradation product, D-dimer, is used as a marker of COVID-19 severity. The hyperinflammatory state from CS often leads to profound pulmonary inflammation and subsequent endothelial damage and platelet aggregation. This thromboinflammation process leads to ARDS in acute COVID-19 and likely contributes to respiratory symptoms of long-COVID [16]. As mentioned earlier, a higher level of TGFβ present in COVID-associated ARDS would contribute to lower levels of GSH in COVID-19 illness [15]. The current management of COVID-19-related ARDS includes ventilatory support, either baricitinib or tocilizumab and dexamethasone [71]. Significantly, the concentration of endogenous GSH in the alveolar epithelial lining fluid of ARDS patients is 5% of comparable healthy individuals [8]. Immunothrombosis biomarkers (specifically levels of tissue factor and vWF) could predict the development of ARDS in moderate-to-severe COVID-19 [60]. Clinically available laboratory markers identifying inflammation, coagulopathy, or tissue injury include neutrophil-lymphocyte ratio, ferritin, C-reactive protein, D-dimer, aspartate aminotransferase, and troponin [67,72,73,74]. Hence, preventing ARDS by avoiding OS-induced cytokine release with early administration of GSH may be an effective strategy to prevent severe consequences of COVID-19 multi-organ injury. 

Ideally, COVID-19 illness should be prevented or attenuated with vaccines. Next, the progression of the illness with severe complications requiring hospitalization should be halted. Finally, organ damage in seriously ill high-risk individuals, particularly respiratory failure from ARDS, which can lead to death, should be prevented. Vaccination usefulness is limited by the constant appearance of new COVID-19 variants, its diminished efficacy in those immune-suppressed, and avoidance by those adverse to vaccines. Once COVID-19 is contracted, Paxlovid is available to high-risk individuals. Although it has been shown to decrease the risk for severe disease progression and may prevent long-COVID, it exhibits drug–drug interactions. It is known to occasionally have a peculiar “rebound” phenomenon of COVID-19 positivity after the recommended 5-day course. Its use has been recently reviewed [75]. Using the GSH-CD complex overcomes many of these limitations and presumably may be administered with or instead of Paxlovid, baricitinib or tocilizumab, and dexamethasone. The complex is an immunomodulator and does not cause immunosuppression. Significantly, the timing and duration of its use is not a critical factor.

Although cyclodextrin (CD) is the transport vehicle in the G/C complex, there is evidence that it may inhibit cellular entry of the SARS-CoV-2 virus. CD may inhibit the viral attachment to the ACE-2 receptor and its subsequent endocytosis entry by altering cholesterol microdomains on the cellular membrane [76].

### 3.3. Long COVID

New, persistent, or relapsing symptoms following an initial acute COVID-19 illness are officially known as post-acute sequelae of COVID-19 (PASC) or, more commonly, as long-COVID (LC). Due to its complex nature, it is not rigorously defined. Still, it is understood to be a constellation of persistent symptoms that occur more than three weeks after the initial acute exposure to the SARS-CoV-2 virus. The outcomes three years after the initial acute infection have been reviewed [77]. The risk of developing PASC is unrelated to the severity of the initial COVID-19 illness. It can occur after mild illness and in those without acute symptoms [78]. The most prominent symptoms include post-exertional malaise (PEM), fatigue, brain fog, dizziness, palpitations, headache, gastrointestinal disturbance, diminished sexual desire or capacity, loss or disturbance in smell or taste, dyspnea, and chronic cough [78,79,80,81]. In addition, there was an increased risk of neuropsychiatric and autoimmune disorders associated with OS [51,81,82,83]. Due to the persistence of these debilitating symptoms, it has become recognized as a second or “rebound” epidemic. It affects the lives and livelihoods of tens of millions of individuals [84]. A review of the presumptive pathophysiologic mechanisms of PASC has included the sequelae of direct viral injury, immune system dysregulation, endothelial dysfunction, mitochondrial metabolic dysregulation, reactivation of harbored pathogens, autoimmunity, and functional changes in blood cells has brought into focus the risks of developing neurologic symptoms similar to myalgic encephalitis/chronic fatigue syndrome (ME/CFS) and post-treatment Lyme disease syndrome (PTLDS) [78,81,85]. Unlike ME/CFS and PTLDS, the sequelae of LC are more complex since they involve the added consequence of tissue injury from acute COVID-19 illness and its influence on exacerbating or precipitating underlying diseases (e.g., diabetes, cardiovascular, and pulmonary) [85]. The similarities and differences between ME/CFS and COVID have been covered in comprehensive reviews [78,85]. Overlapping symptoms with ME/CFS include fatigue, PEM, headaches, impaired reasoning or “brain fog”, myalgia/arthralgia, orthostatic intolerance, nausea/diarrhea, and cough [79]. Symptoms peculiar to long-COVID include decreased or distorted smell and taste, rash, and hair loss. Those peculiar to ME/CFS include painful lymph nodes and tinnitus [85]. Likewise, a comparison of ME/CFS to PTLDS has been presented [86]. There is evidence to support that they all follow a trigger of autoimmunity that stems from the effects of persistent T-cell activation, and in COVID, may additionally reflect the antigenic stimulation by residual intact viruses or viral remnants [72,82,87,88]. 

Further elucidation of the presumptive mechanism of long-COVID was reported in an exhaustive prospective clinical analysis of 113 COVID-19 vs. 39 healthy controls followed for up to a year after initial confirmation of acute SARS-CoV-2 illness. Over 6500 biomarkers were analyzed. This study showed that long-COVID patients exhibited an enhanced complement activation during the acute illness that persisted at 6-month follow-up [57]. Furthermore, long-COVID patients had evidence of persistent thromboinflammation with its characteristic endothelial activation of vWF and thrombin cascade, platelet aggregation, and prolonged T-cell activation [61,87]. Finally, these patients had an associated increased IgG antibody level against S protein, suggesting a predisposing risk factor. Similar to ME/CFS, the presence of cytomegalovirus or Epstein–Barr virus antibodies is a possible risk factor of long-COVID [89]. Significantly, the role of GSH in inhibiting antibody and complement-mediated immunologic cellular injury has been presented [90].

## 4. Diabetes and Its Complications

### 4.1. Diabetes Management

The treatment goals of diabetes management are to avoid the personal and healthcare burden of its neurovascular complications. These complications result from hyperglycemia (both acute and chronic) that leads to OS and the subsequent interconnected inflammation, endothelial dysfunction, and hypercoaguability [91,92,93,94]. The evolving view is that diabetes is a metabolic disease resulting from chronic inflammation driven by OS [94,95,96]. This chronic inflammation has been implicated in the development of insulin resistance and diabetes complications (DC) [97,98]. The pathological basis of DC has historically been dichotomized as micro- or macro-vascular. These include microvascular (retinopathy, nephropathy, and neuropathy) and macrovascular (coronary heart disease and stroke). A more holistic concept of the disease processes as “panvascular” was proposed. It recognized that micro- and macro-vascular pathologies often co-exist and are interactive [94].

The current target of optimal glucose control is an A1c of <7, which is higher than the A1c of <5.4 of non-diabetic individuals. The higher A1c treatment goal is to avoid severe and potentially fatal episodes of hypoglycemia. Continuous glucose monitoring has reduced this risk for those using insulin. Episodes of hypoglycemia in type 2 diabetes have been reduced by the substitution of sulfonylurea/meglitinide medications with newer agents (e.g., dipeptidyl peptidase four inhibitors (DPP4i), glucagon-like peptide-1 receptor agonists (GLP1ra), and sodium/glucose transporter two inhibitors (SGLT2i)). However, the goal of normalizing A1c is often difficult to attain—primarily due to insulin resistance (IR).

OS has been implicated in the development of IR and β-cell dysfunction [97,99,100]. Of note, oscillating glycemic levels from high post-prandial glucose from IGT are strongly linked to both β-cell dysfunction, IR, and cardiovascular disease (CVD) [101]. IR itself is an independent risk factor for CVD that is associated with mitochondrial dysfunction [93]. From the longitudinal studies on diabetes care, it was determined that once diabetes is established, A1c control alone may not prevent complications of diabetes due to long-lasting epigenetic changes that drive persistent expression of pro-inflammatory genes after glycemia is normalized, known as “hyperglycemic memory” [93].

Finally, the metabolic role of fumarate in diabetes and its attack on the sulfydryl groups in mitochondria to form succinate should be noted [102]. GSH is one target of the process. Recently, the association of succinate (a signaling metabolite) with the gut microbiome, obesity, diabetes, cardiovascular diseases, and nonalcoholic fatty liver disease (NAFLD) was reviewed. They further propose that succinate may be a marker of inflammation in metabolic disorders, particularly type 2 diabetes and its complications [103].

### 4.2. Diabetes Complications

Without the normalization of hyperglycemia, protein glycation leads to the formation of advanced glycation end-products (AGEs), which induces OS in a positive feedback loop. AGEs are heterogeneous molecules formed from irreversible, non-enzymatic glycation of proteins, lipids, and nucleic acids [93,99,104,105].

AGEs contribute significantly to the vasculopathies associated with diabetes. On a macrovascular level, this includes impaired vasculogenesis in response to ischemia, resulting in defective generation of collateral blood flow. This impaired adaptive response increases the risk of limb amputations and myocardial infarction [93]. Upon binding to receptors for AGEs (RAGEs), an inflammatory and pro-coagulant state is initiated, which causes endothelial activation. The resultant expression of inflammatory cytokines and adhesion molecules amplify the inflammatory response. Among the cytokines are TNF-a, IL-1, and IL-6, which stimulate the release of pro-coagulant molecules and inhibit the expression of anti-coagulant molecules. This leads to microvascular disturbances, including lipid deposition, sclerosis, and impaired vasodilation. Similar to the processes of ARDS in COVID-19, there is an activation of vWF from damaged endothelial cells that stimulates platelet aggregation [92]. Significantly, high levels of vWF in patients with diabetes appear to be a predictive biomarker of diabetic nephropathy. A more detailed treatise of the role of AGEs in causing the various neuro-vascular diabetes complications has been presented [104]. The authors reviewed the role of the transcription factor—nuclear factor kappa B (NF-κB) in triggering the development of OS, cytokines and chemokines, cell adhesion molecules, interleukins, tissue growth factor (TGF), pro-inflammatory proteins, and pro-apoptotic genes [104].

The definition of metabolic syndrome (MetS) and its CVD risks has been outlined [106,107]. These include insulin resistance, obesity, abnormal lipid profile, hypertension, and impaired glucose tolerance. Clinical criteria for diagnosing MetS were also presented. Furthermore, evidence has been observed that OS links IR and the pathology associated with the MetS [95]. These include OS due to lipid peroxidation, AGE formation, and decreased antioxidant protection, resulting in CVD complications from monocyte activation, vascular smooth muscle proliferation, and endothelial cell dysfunction. The latter additionally leads to arterial hypertension and plaque formation [95]. 

MetS is associated with NAFLD and its inflammatory form, nonalcoholic steatohepatitis (NASH). NASH appears to follow the “two hit” phenomenon where the initial hepatic lipid infiltration is followed by OS, which leads to tissue injury and subsequent fibrosis, which is associated with enhanced lipid peroxidation [95].

Analogous to the concept expressed by Grattagliano, a detailed presentation by Giacco hypothesizes that OS plays a pivotal role in micro- and macro-vascular diabetes complications [93,95]. ROS, particularly superoxide production by mitochondria, is involved in the pathogenesis of polyneuropathic and nephropathic complications [93,108,109]. These include activation of five major pathways: increased flux of glucose and other sugars via the polyol pathway, increased formation of AGEs, increased expression of RAGE and its activating ligands, activation of protein kinase C (PKC) isoforms, and overactivity of the hexosamine pathway. It has been summarized as a being generated by “glycoxidation”, a combination of glycation and oxidation driven by OS [105]. Microvascular pathology is a consequence of intracellular hyperglycemia, and macrovascular pathology is caused by increased oxidation of fatty acids [93]. It appears that a common upstream event triggers all five mechanisms—hypothetically, OS from overproduction of ROS by mitochondria. Significantly, in experimental animal models, treatment with agents directly opposing SOD normalized all five pathways. GSH, as being the most essential and abundant low molecular intracellular ROS scavenger, has been noted [105]. The efficacy of exogenous administration of GSH with the GSH/CD complex remains to be determined.

The role of OS in diabetic nephropathy, neuropathy, and retinopathy has been reviewed [94,108,110,111,112,113]. Presumably, if GSH can prevent the pathologic consequence of glycation resulting from OS, it can be used in conjunction with glucose control to prevent diabetes complications. In essence, it is used to simultaneously treat the cause and effect of hyperglycemia [94]. The dose and cost of insulin and oral medications could be reduced by addressing insulin resistance. Prevention or a delay in the onset of diabetes complications could be achieved by preventing protein glycation. The incidence of neurovascular complications, including macrovascular (stroke, myocardial infarction) and microvascular (blindness, chronic kidney disease) should decline. The socio-economic impact of reducing the burden of caring for individuals with diabetes complications and the avoidance of their impaired quality of life would be substantial. OS plays a critical role in the progression of diabetes complications. What is lacking is an effective antioxidant to address it [96,100,114,115].

## 5. Neurologic and Neurodegenerative Disorders

The healthcare impact of neurodegenerative disorders cannot be understated. The burden of Alzheimer’s disease (AD) and related dementias (including AD, vascular dementia, Lewey body dementia, and frontotemporal dementia) is present in an estimated 6.5 million individuals in the US. This incidence is projected to increase to 13.8 million individuals by 2060 [116]. GSH deficiency and OS have been associated with many degenerative neurologic illnesses, including AD, Parkinson’s disease (PD), amyotrophic lateral sclerosis (ALS), Huntington’s disease (HD), multiple sclerosis (MS), and autism spectrum disorders (ASDs) [117,118,119,120,121,122,123,124,125,126,127]. OS has also been associated with cognitive impairment of diabetes and advanced age [121,128,129]. Aoyama et. al. reviewed the association of neural GSH deficiency with Alzheimer’s disease (AD), Parkinson’s disease (PD), and (ALS) [117]. Algahtani et al., described the association of mitochondrial dysfunction and OS in AD, PD, HD, and ALS [130]. Underscored was the role of GSH in counterbalancing OS. Using proton magnetic resonance spectroscopy (^1^H-MRS) to detect GSH levels in vivo, evidence was presented of GSH deficiency in the brains of AD, PD, and ALS individuals. Expressly, a lower level of GSH was noted in the temporal and parietal lobes of older adults, with a negative correlation of GSH levels to Aβ levels as assessed with positron-emission tomography (PET) imaging. In PD, there is post-mortem evidence of decreased GSH levels in the substantia nigra of the midbrain. Finally, again, using ^1^H-MRS, a decrease in GSH levels was noted in the motor cortex and corticospinal tract of individuals with ALS [117].

### 5.1. Alzheimer’s Disease

Alzheimer’s disease is a neurodegenerative disorder with neuropathologic hallmarks that include amyloid beta (Aβ) plaques, neurofibrillary tangles (NFTs), synapse and neuronal loss, neuroinflammation, neurotransmitter deficiencies, and reactive astrogliosis, which eventually lead to cognitive impairment [131]. Although classical AD comprises 85% of cases, there are other phenotypes, including the lipogenic variant of primary progressive aphasia, posterior cortical atrophy, corticobasal syndrome, and frontal AD [131]. There is growing evidence of the involvement of mitochondrial dysfunction and OS early in the disease progression, including elevated levels of OS markers [132,133]. This includes increased malondialdehyde (MDA), a lipid peroxidation product, early in AD [134]. The treatment of Alzheimer’s disease (AD) with disease-modifying therapies (DMTs) has been disappointing [135,136,137]. In addition to neurotransmitter enhancement, disease management has been mainly directed against Aβ deposition and NFT formation. This approach may address AD at a point of the disease process that is difficult to modify or reverse. That is, Ab plaques and NFT formation may be the result and not the cause of AD pathology [138]. Rather than treating AD-associated damage to neurons, avoiding the pathologic insults of Aβ and neurofibrillary tangles may be more effective by preventing their formation. Cheong et al. highlighted the complex and multifactorial disease hypotheses (amyloid cascade, tau formation, and cholinergic dysfunction) that direct current therapeutic interventions. DMT treatments are directed against Ab and tau, and memory deficits are addressed with neurotransmitter therapy [139].

Current established treatments address neurotransmitter imbalance. Anti-amyloid therapies have been disappointing. It has been proposed that this approach may be too little and too late [135,136,140]. This may require addressing AD early, at its pre-clinical stage, preceding mild cognitive impairment directed against OS [38,138,141,142,143]. The obvious barrier to this approach is having methods to identify those at-risk individuals before symptoms of dementia are evident [131]. Encouragingly, early markers and imaging methods have been identified. These include PET scans, functional and proton MRI, AD associate gene markers, ocular imaging, and MDA (a biomarker of lipid peroxidation) [117,126,131,144,145,146,147,148,149,150]. The measurement of a particular combination of CSF biomarkers has a sensitivity of 95% and specificity of 83% and is used in clinical trials to detect individuals who will develop AD [135]. Unfortunately, these interventions are still research tools and are not yet available in routine clinical practice due to their cost, access, or invasiveness. Ocular examination as a non-invasive method for early AD detection has been proposed [146,151,152,153]. OS has been recognized as the initial and driving force in Aβ deposition, and Aβ, in turn, may be a compensatory response to OS [134,151,154]. The scientific conundrum of which came first, Aβ or OS, would be a moot issue if early reversal of OS prevents Aβ deposition and neurofibrillary tangle formation before their burden causes clinical symptoms [38,133,155]. The pathophysiology of AD is complex, is not fully understood, and warrants the studies of alternate disease mechanisms and the use of combination therapies [138,139,150,156,157,158,159,160,161]. 

In a detailed review, Walker et al. described in detail the role of peripheral inflammatory insults that increase the future risk of developing Alzheimer’s disease [162]. The insults that cause peripheral inflammation could be an acute infection (e.g., COVID-19, pneumonia, septicemia) or a sterile event (e.g., surgery, orthopedic injury, acute coronary syndrome). If the insult leads to the complication of hypoxia, it will trigger additional IRI-associated inflammation. Another risk factor is age-related immunosenescence, which leads to the upregulation of inflammation. The group further describes the subsequent deterioration of the blood–brain barrier (BBB) that would allow crosstalk of the peripheral with the central nervous system inflammatory events. They propose that Alzheimer’s disease can be a result of an acute or chronic inflammatory process. Acutely by triggering innate immune activation and chronically through chronic cytokine expression [162]. Through several meetings of the Alzheimer’s Association Research Roundtable, it was affirmed that clinically meaningful treatments need to either slow the decline or prevent future impairment [163]. The current disappointing outcomes of treatment to slow the decline may prompt a greater focus on its prevention. Nevertheless, providing medication to prevent the onset of dementia raises not only the medico-economic challenge of establishing an accurate and cost-effective method of identifying future dementia risk but also the determining the cost vs. benefit of treating an asymptomatic individual [162,164,165].

### 5.2. Parkinson’s Disease

After AD, PD is the second most common neurodegenerative disorder. Its symptoms are attributed to the selective degeneration of dopamine (DA) neurons in the substantia nigra (SN). It has been proposed that chronic OS drives the pathogenesis of the disease [166,167]. Individuals with PD have higher serum levels of OS markers with a simultaneous low level of GSH in the SN that enhances PD vulnerability [168]. Mitochondrial injury from 1-methyl-4-phenyl-1,2,3,6-tetrahydropyridine (MPTP) exposure produces OS. MPTP was initially an illicit drug that produced PD in its users. It is now used experimentally in animal studies to induce PD. Key evidence of the significance of OS in PD was (1) an increased ROS and heightened OS driven by endoplasmic reticulum (ER) stress, (2) alpha-synuclein (α-SN) accumulation that is affected by and contributes to OS, and (3) that extracellular DA itself promotes oxidation and increases the vulnerability to OS of DA-producing neurons in the SN, thus concluding that OS is the common denominator driving the development of PD [166]. PD demonstrates the loss of dopaminergic neurons accompanied by OS and preceded by GSH depletion [145]. Interestingly, OS may have a critical role in the pathogenesis of Huntington’s disease, amyotrophic lateral sclerosis, and multiple sclerosis [122,130,169,170,171]. 

### 5.3. Autism Spectrum Disorder

Autism spectrum disorder (ASD) is a clinically diagnosed developmental disorder usually determined at 4.5 years of age. It is characterized by impaired social communication and interactions, repetitive behavior, and limited interests or activities. Its etiology and pathogenesis are poorly understood. There are no accepted biomarkers for ASD. It has been proposed that OS (and associated GSH deficiency) is a vital and integral factor in a multifaceted pathological process [172,173,174,175,176,177,178,179]. The level of GSH may be monitored in the serum by a ratio of reduced GHS to its oxidized dimer GSSG (the GSH:GSSG ratio) and with ^1^H-MRI [172,177,179,180]. Serum markers of OS include elevated homocysteine level (and its associated folate and cyanocobalamin deficiency), lipid peroxidation with elevated MDA level, numerous inflammatory cytokines (e.g., interferon-gamma (IFN-γ), interleukin 6 (Il-6), and tumor necrosis factor (TNF) [173,174]. Current treatments address the symptoms of ASD but not the underlying pathophysiology that may be a result of OS [175,181]. Hopefully, studies using the G/C complex will provide a definitive assessment of the influence of OS and GSH in ASD.

### 5.4. Encephalopathy

Evidence was presented that OS in Wernicke’s encephalopathy may be addressed by administering antioxidants in addition to thiamine supplementation [182]. In addition, OS appears to play a critical role in the pathogenesis and progression of hypoxic-ischemic encephalopathy in neonates [183]. 

The pathophysiology of neurologic diseases is complex, and the significance of OS in modifying its development remains to be established. Gribkoff reviewed possible reasons for current treatment failures, including (1) ineffective transport across the blood–brain barrier (BBB), (2) finding animal models that recreate the corresponding human disease, and (3) addressing biomarkers that may not influence the disease [159]. Despite recognizing the utility of GSH in addressing OS, a lack of an effective delivery method has hindered its use in clinical studies [174]. The oral administration of GSH faces its degradation by gastrointestinal peptidase. Intravenous GSH is rapidly metabolized with an elimination half-life of around 7 min. (Ayoama) Transporting GSH across the blood–brain barrier (BBB) has also been an issue for brain diseases. Significantly, it has been demonstrated that γ-cyclodextrin crosses the BBB [64].

## 6. Fibrosis 

Fibrosis is a pathologic consequence of chronic inflammation that involves multiple organs (heart, lungs, liver, kidneys, skin) [165]. It is a sequela of tissue injury that leads to an excessive buildup of fibrous connective tissue in the extracellular matrix. Among the illnesses that cause chronic inflammation leading to fibrosis include organ transplantation, high cholesterol, myocardial infarction, idiopathic pulmonary fibrosis, poorly controlled diabetes, obesity, and hypertension [63,184]. The fibrotic process is triggered and mediated by transforming growth factor beta (TGF-β). Other mediators include fibroblast growth factor (FGF), connective tissue growth factor (CTGF), nuclear erythroid 2-related factor 2 (nrf2), and the renin-angiotensin-aldosterone system. (RAAS) Finally, the critical role of OS in the pathogenesis of fibrosis and its interaction with TGF-β in a positive feedback loop is detailed by Antar et al. [184]. Chronic inflammation is strongly implicated in fibrosis that leads to diabetes complications, renal fibrosis, pulmonary fibrosis, hepatic cirrhosis, and cardiac remodeling [184]. The role of innate and adaptive immune cells in the fibrotic process has also been covered. In particular, a detailed description of the influence of cytokines produced by adaptive immune cells in the pathogenesis of fibrosis was presented [185]. An exhaustive literature review on OS in pulmonary diseases highlighted the top thirty drugs used in 31,373 publications [63]. The most studied drug was NAC. GSH was not on the list, likely reflecting the current absence of an effectively administered compound.

### 6.1. Pulmonary Fibrosis

The main parenchyma of the lung is comprised of alveolar epithelial cells (AECs)—formerly known as pneumocytes. Of the two subtypes, AEC1 forms the alveoli where gas exchange occurs. ACE2, adjacent to AEC1, has a more complex role. It supplies a surfactant that maintains alveolar integrity, transforms into AEC1 in adaptive tissue repair, and is involved in the inflammatory response. Sustained insults result in a maladaptive response of AEC2 to recruit myofibroblasts, activation of collagen synthesis, AEC2 apoptosis, and ultimately pulmonary fibrosis (PF) [43,45]. This process is stimulated by pro-inflammatory cytokines, particularly tissue growth factor beta (TGF-β), which increases alveolar secretion of fibroblast growth factor 2 (FGF-2). Other cytokines include tumor necrosis factor-alpha (TNF-α) and platelet-derived growth factor (PDGF). OS appears to be implicated in the fibrotic process by inducing epithelial apoptosis, increasing cytokine levels, and promoting the differentiation of fibroblasts to myofibroblasts. Effective prevention of PF should target early intervention of the immunothrombotic/fibrotic process and the development of pulmonary fibrosis associated with the depletion of GSH [45,46]. Murine animal studies with inhaled NAC appear to attenuate bleomycin-induced PF [33]. However, human studies with NAC have been disappointing [46]. Prior studies with antioxidants have shown either toxicity or lack of efficacy in human subjects. Most of the studied compounds are natural flavonoids or polyphenols that have demonstrated poor efficacy in humans. It could be due to an intrinsic lack of efficacy, inadequate dosage, or inadequate duration of use [46]. However, none have used GSH via the G/C complex. Finally, like addressing dementia, the role of treatment may be more effective in preventing their onset and progression rather than reversing the pathological processes of these illnesses.

### 6.2. Cystic Fibrosis

Another chronic, progressive lung disease associated with high OS is cystic fibrosis (CF) [186]. It is a disease that occurs due to mutations in the gene encoding for the CF transmembrane conductance regulator (CFTR). The primary function of CFTR is to regulate the efflux of chloride and bicarbonate anions. Its dysfunction leads to a water-electrolyte imbalance on the surface of numerous organs and organ systems, including the upper and lower airways, intestine, skeletal muscles, pancreas, biliary tree, cervix, vas deferens, and sweat glands. Clinically, it results in inflammation, recurrent airway infection, and a progressive decline in lung function leading to death [48,187]. Recurrent pulmonary infections lead to a self-amplifying process of cellular damage that involves OS. Given the complexity and variability of CF, it was proposed that the disease be managed with a combination of antioxidant and anti-inflammatory agents plus CFTR-targeted therapies to address the interrelationship of CFTR and OS [48]. It may be due to an inability to increase GSH in the epithelial lining fluid (due to the deficiency of CFTR), which may contribute to the poor response to infection in CF patients [9].

### 6.3. Hepatic Cirrhosis

Hepatic cirrhosis is a major worldwide health problem. Its etiologies include alcohol abuse, hepatitis B and C infection, and nonalcoholic fatty liver disease (NAFLD). With the success of vaccination against hepatitis B and the treatment of hepatitis C, the risk of developing cirrhosis from NAFLD is becoming a significant problem. It is the most common cause of liver disease worldwide and is influenced by the growing epidemic of obesity and its associated MetS [188]. The incidence of NAFLD, which is up to 75% in obese individuals and even higher in those with T2DM, has been noted [50]. As reported in 2018, NASH-related cirrhosis accounted for 5% of all young patients listed for liver transplantation in the United States [188]. Finally, the risks of complications (hepatorenal syndrome, variceal hemorrhage, hepatic encephalopathy, and hepatocellular carcinoma) from NASH and hepatitis C cirrhosis were similar. [188] 

Although the pathway from NAFLD to cirrhosis is not firmly established, there is strong evidence of OS influence in its process. This association should not be surprising, given the role of OS in IR and MetS, as mentioned earlier. It has been proposed that hepatic dysfunction is the expression of MetS in the liver [50]. This process is initiated simultaneously by fat infiltration of hepatocytes, IR, and mitochondrial beta-oxidation, leading to increased ROS and eventual OS due to mitochondrial dysfunction, ER stress, derangement of iron metabolism, dysbiosis in the gut (including heightened ethanol-producing bacteria), and endothelial dysfunction (ED) [50,189]. As described by Masarone et al., ED is a result of excessive production of nitric oxide (NO), which in turn contributes to and promotes the structural and functional changes of liver circulation, impairs regeneration after liver injury, and thereby leads to the progression of liver disease [50]. This is a modification of Day and James’s “two-hit” hypothesis. They proposed that steatosis is the first “hit” that requires modifying factors (primarily linked to OS) as a second “hit” to produce the fibrosis and cirrhosis resulting from nonalcoholic steatohepatitis (NASH) [50]. The pathologic sequence was summarized as a consequence of (1) excessive fatty acids in hepatocytes, (2) their energy depletion and resultant mitochondrial dysfunction, (3) an increase in OS, and finally, (4) cellular damage. The association of concurrent decreased antioxidants, particularly of GSH depletion, was also noted [49]. The overall complexity of the pathogenesis of NASH-cirrhosis was diagramed by Bovi—Figure 1. Modest benefits after four months with the administration of a modified oral GSH supplement have been reported [49,189]. 

The importance of OS in the pathogenesis of cirrhosis in hereditary hemochromatosis (HHC), chronic hepatitis C virus infection, and primary biliary cirrhosis has been reviewed [190,191]. It has been proposed that early interventions against, including maintaining GSH levels, may protect hepatocytes from damage [172]. Notably, the presence of cirrhosis is associated with a lower-than-normal life expectancy and a significantly increased risk for the development of hepatocellular carcinoma.

## 7. Cardiovascular Disease

Cardiovascular disease (CVD) is a group of disorders of the heart and blood vessels. The interrelated disease entities described are atherosclerosis heart disease (ASHD), coronary artery disease (CAD), arterial hypertension (AH), and restenosis [53,192,193]. ASHD is a chronic progressive thickening, hardening, and narrowing of the systemic arterial walls primarily due to elevated plasma cholesterol. CAD is a narrowing of coronary arteries primarily caused by the formation of unstable plaques within the intima of the vessel wall. A rupture of unstable plaques results in acute thrombosis, which is a significant cause of acute myocardial infarction and strokes. The role of the immunothrombotic process leading to thromboinflamation was discussed previously in hepatic and COVID-related coagulopathies—specifically, the cyclic role of the coagulation cascade involving platelet and innate immune cell activation due to cytokine dysregulation from OS [10,33,40,194]. The role of OS from SARS-CoV-2 infection and its subsequent pro-atherogenic CVD complications (heart failure, stroke, venous thrombosis, and pulmonary embolism) after COVID-19 illness has been presented [41,195,196]. The current lack of an effective antioxidant therapy was noted [194]. Finally, the role of ROS generation by statins in producing tissue toxicity of the liver and muscles was discussed [10]. 

## 8. Viral Illnesses

### 8.1. Encephalitis and Thrombosis

The pathologic consequence of viral illness from SARS-CoV-2 and hepatitis C has been reviewed. The role of OS infections from the Flaviviridae family was discussed [197]. It highlighted processes by which OS facilitates viral replication and the pathologic consequences of the resultant illnesses. These viruses include dengue virus (DENV), Zika virus (ZIKV), yellow fever virus (YFV), Japanese encephalitis virus (JEV), West Nile virus (WNV), tick-borne encephalitis virus (TBEV), and hepatitis C virus (HCV). The major clinical manifestations are hemorrhagic (DENV, YFV), neurologic (ZIKV, JEV, WNV), and hepatic (HCV) symptoms. Growing evidence supports the link between OS and viral pathogenesis—particularly HCV and DENV and the respiratory syncytial virus (RSV) [197,198]. Although direct-acting antiviral agents (DAAs) against HCV have significantly avoided its serious complications, the high cost and limited availability of DAA keep it beyond the reach of many individuals—particularly those in Third World countries. Analogous to SARS-CoV-2 infection, Ebola and Hantavirus appear to cause a cytokine storm that leads to pulmonary and renal injury [199,200]. The role of GSH treatment in these viral illnesses remains to be determined.

In addition, the role of neutrophil extracellular trap (NET) formation and complement activation in response to bacterial and virus infections is significant. NETs and complement significantly contribute to thrombosis, atherosclerosis, and autoimmune diseases, and they have been described in detail [16,33].

### 8.2. Myocarditis

The pathology of viral cardiomyopathy is complex. A thorough review of virus-induced myocarditis and cardiomyopathy has been presented [201]. The distinction between the direct, immediate virus-induced (cardiotropic) cytotoxicity and the indirect, delayed virus-associated, immune-mediated (vasculopropic and lymphotropic) inflammatory cardiomyopathy has been discussed [201]. The numerous viruses include adenoviruses, enteroviruses (e.g., coxsackievirus), parvovirus B19, cytomegalovirus, Epstein–Barr virus, hepatitis C, human immunodeficiency virus (HIV), and coronavirus (including SARS-CoV-2). The role of the innate and adaptive immune response, the neutrophil, eosinophil, T and B cell response, and autoimmunity has also been reviewed. In this context, the influence and treatment of OS with GSH also remain to be determined.

### 8.3. Hemorrhagic Fever

Many enveloped RNA viruses cause a syndrome known as viral hemorrhagic fever (VHF) that is driven by cytokine dysregulation [202,203,204]. These include the Ebola virus (EBOV), the Marburg virus (MARV), the Lassa virus (LAV), and the previously mentioned YFV and DENV. Until a successful vaccine or antiviral agent against these viruses is developed, the current treatment is supportive care. In the interim, addressing the cytokine dysregulation with the G/C complex will hopefully reduce the incidence of mortality and severe morbidity from those viruses. 

### 8.4. Pneumonia

Respiratory syncytial virus (RSV) is a significant cause of pneumonia in children. Infants experience the most severe complications requiring hospitalization. Although a vaccine against RSV is available, current treatment for those contracting the illness is supportive. The pathogenesis of serious illness appears due to the inflammatory response of the lungs to OS. Markers of oxidative damage correlated with the severity of damage [205,206]. Since the G/C complex is applied topically, future studies on infants and neonates will not require typically difficult IV access.

## 9. Ischemia-Reperfusion Injury

The role of OS in ischemia-reperfusion injury (IRI) is another area for future studies. These include sequelae of myocardial infarction (MI), stroke (ischemic and hemorrhagic), shock, and organ transplantation. It has become recognized that organ injury occurs not only as a result of initial ischemia but of subsequent reperfusion [207,208,209]. Reperfusion injury occurs paradoxically upon reoxygenation with the restoration of blood flow. Cellular dysfunction and death result from OS and its associated inflammatory response [209]. The inflammatory response is necessary for tissue repair and includes neutrophil formation, cytokine production (including TNF-α, IL-1β, and IL-6), and other pro-inflammatory stimuli [207]. OS can usually be identified by measuring malondialdehyde (MDA), a byproduct of OS of lipid peroxidation. The IRI process may involve not only the initial organ but also cause systemic damage to distant organs and potentially lead to multi-system organ failure [209].

### 9.1. Myocardial Ischemia

Prompt reperfusion after acute myocardial infarction (AMI) is a critical cardiovascular intervention. Nevertheless, the subsequent reperfusion may lead to dysfunction more severe than the initial ischemia. These include myocardial stunning, reperfusion arrhythmia, myocyte death, endothelial and microvascular dysfunction that includes the no-reflow phenomenon, and an inflammatory response. Detailed reviews of the process of cardiomyocyte death that follows mitochondrial dysfunction, calcium overload, and a maladaptive inflammatory response that OS drives have been presented [210,211,212]. A prospective clinical study of OS in myocardial infarction-reperfusion injury (MIRI) confirmed the diagnostic value of OS markers. The authors compared levels of superoxide dismutase (SOD), malondialdehyde (MDA), myeloperoxidase (MPO), ischemia-modified albumin (IMA), and heat shock protein 70 (HSP70) after reperfusion to percutaneous intervention (PCI) in elderly patients with AMI and type 2 diabetes [213]. The presence of those markers predicted a higher risk of IRI injury. Presumably, measures to prevent or minimize OS could result in more favorable clinical outcomes [214].

### 9.2. Central Nervous System Injuries

The IRI injuries following OS from acute ischemic stroke (IS) have been reviewed [215,216,217]. OS drives the initial endothelial dysfunction in IS with subsequent platelet aggregation and a coagulation cascade initiated by activated vWF [216]. Successful recanalization procedures, with prompt cerebral blood flow reestablishment, have improved patient outcomes. Paradoxically, reperfusion may worsen neurologic function through cerebral edema and thrombosis [215]. The critical role of OS and subsequent mitochondrial dysfunction leading to apoptosis in potentially salvageable ischemic surrounding areas was also noted. 

Additionally, OS appears to have a significant role in the secondary outcomes of the initial injury from intracerebral hemorrhage (ICH), which is similar to IS. Following the initial insult from the mass effect of the hematoma, the secondary brain injury (SBI) is attributed to neuroinflammation, excitatory amino acid, cytotoxicity of blood, hypermetabolism, and spreading depression driven by OS [218,219]. The brain is highly susceptible to OS damage since it is rich in lipids and iron but is deficient in antioxidants. The interplay of OS and the secondary outcomes of ICH (neuroinflammation, glutamate excitotoxicity, cell death, and disruption of the blood–brain barrier (BBB) has been reviewed [216,218]. A higher level of OS, usually measured with MDA, is associated with more severe clinical harm. Current interventions include treating the cause or source of the bleed and reducing its hemorrhagic load with either a surgical or minimally invasive decompression procedure [220]. Addressing SBI should further improve the current neurologic outcomes after ICH.

### 9.3. Organ Transplantation

IRI is an inevitable consequence of organ transplantation. Effective organ preservation and successful transplantation is critical since the need for organs continues to exceed its supply. By reducing OS, increased donor pool and transplantation success have benefitted from machine perfusion (MP) technology [221]. IRI caused by OS during early reperfusion negatively affects both short- and long-term outcomes following transplantation [222]. This reflects the significance of mitochondrial dysfunction and immunothrombosis produced by ischemia and reperfusion [208]. Evidence has been summarized that OS-induced IRI is a crucial event during the full spectrum of transplantation from retrieval, preservation, and reperfusion [222].

### 9.4. Sickle Cell Disease

Sickle cell disease (SCD) is a global health burden affecting over 40 million individuals with the disease or trait. Of those, 3.2 million live with the disease, a vaso-occlusive consequence of impaired blood rheology, hemolysis, increased adhesiveness of erythrocytes to vascular endothelium with co-existing inflammation, and hemostasis activation [223,224,225]. OS and its subsequent inflammatory response are driven by IRI through the Fenton reaction resulting from iron release due to chronic hemolysis [223,225,226]. Similar to diabetes, AGEs appear to have a role in the pathology of SCD [226]. Likewise, the similarity of thomboinflammation in COVID-19 and SCD has been noted [227]. Finally, advanced lipoxidation end-products (ALEs) and lipid peroxidation (as measured with high levels of MDA) have been identified [226]. A low level of GSH in RBCs and the impairment of its de novo synthesis has been recognized [226]. The administration of GSH directly via the G/C complex may overcome these deficiencies. As a relatively inexpensive topical product, the complex would be readily available for chronic use in rural and undeveloped regions.

## 10. Pathologic Aging

Pathologic premature aging, or progeria, is a group of rare inherited genetic disorders associated with OS [228,229,230]. These include Hutchinson–Gilford progeria syndrome (HGPS) and the Werner syndrome (WS). Although the underlying molecular mechanisms of disease differ, they both lead to premature pathologic aging. HGPS is characterized by accelerated aging with cardiovascular dysfunction that leads to death at an average age of 13 years, usually from myocardial infarction or stroke. Unlike HGPS and atypical WS, the onset of symptoms in classic WS occurs after puberty [231]. Although there are several hallmarks of progeria (including telomere dysfunction, increased DNA damage, cell cycle dysregulation, and stem cell exhaustion), inflammaging appears to play an integral role [228]. Antioxidants are widely promoted and used for anti-aging purposes [6]. The use of antioxidants in pathologic premature aging has not been well studied [5]. The effect of early intervention with antioxidants, including GSH, remains to be determined [232].

## 11. Toxicities

In addition to illnesses, OS is induced by toxic substances—both from an overdose and as a side effect. A well-known overdose is from acetaminophen (also known as paracetamol) [233]. Toxicities can occur from exposure to heavy metals (mainly lead and mercury), contrast agents, and some chemotherapeutic agents (anthracyclines, bleomycin, and cisplatin) [234,235,236]. 

### 11.1. Chemotherapy

Chemotherapeutic toxicity can occur acutely (during the treatment course) or delayed (several years after administration). These adverse effects are associated with OS, including anthracyclines, bleomycin, and cisplatin [234]. Acute renal toxicity of cisplatin is complex [237]. The current reno-protective protocol includes saline hydration and mannitol administration [238]. The cardioprotective effect against anthracyclines with enalapril and carvedilol was effective if initiated within two months of completion of chemotherapy but not if initiated after six months [239]. Carrasco et al. suggested the use of antioxidants against anthracycline-induced cardiotoxicity [240]. However, addressing OS from chemotherapeutic agents must be balanced between their antineoplastic activity and their toxic adverse side effects; hence, the use and timing of antioxidants remain to be established [235,238,240,241,242]. Finally, treating chemotherapy-associated toxicity is complicated by the fact that most cancer treatments involve a “cocktail” of two or more agents.

### 11.2. Antiretroviral Therapy

The possible role of OS in toxic adverse effects from highly active antiretroviral therapy (HAART) for human immunodeficiency virus (HIV) infections has been presented [10]. The toxic effects on the liver from mitochondrial dysfunction (steatosis, steatohepatitis, and disorders of lipid regulation) were discussed.

### 11.3. Contrast-Induced Nephropathy

Contrast-induced nephropathy (CIN) is multifaceted and includes direct tubular toxicity, intrarenal vasoconstriction, and excessive production of ROS [236]. Although CIN usually resolves, 15% may require temporary dialysis and 4% progress to end-stage renal disease. High-risk individuals include those with prior chronic kidney disease, congestive heart failure, volume depletion, advanced age, hypertension, and hyperuricemia. In vitro and animal studies have supported using antioxidants for CIN prevention [236]. The clinical use of GSH to prevent CIN remains to be investigated

### 11.4. Acetaminophen Hepatotoxicity

Acetaminophen (APAP) hepatotoxicity is the most common cause of acute liver failure in the United States, with unintentional overdose (OD) comprising the majority of episodes. APAP is typically eliminated through conjugation through glucuronidation and sulfonation. A secondary, hepatotoxic metabolite (N-acetyl-*p*-benzoquinone imine (NAPQI) is normally rendered harmless by conjugation with GSH. However, when the level of APAP overwhelms the primary glucuronidation pathway, the GSH stores are consumed by a generation of the excess NAPQI metabolites and by the formation of mitochondrial NAPQI protein adducts, leading to hepatocellular injury [243]. The standard treatment for APAP overdose is the use of NAC. Early administration of NAC is critical [233]. While this is possible with intentional OD, it is not practical with unintentional OD. The latter instance is usually associated with the overuse of narcotic/APAP pain medications [233]. For intentional OD, direct administration of GSH using the G/C complex would overcome the inherent delay NAC requires to promote de novo GSH synthesis and extend the ability to prevent hepatotoxicity beyond the current 72 h window. Furthermore, the concurrent administration of the complex when APAP use is excessive may prevent unintentional OD

### 11.5. Isoniazid and Rifampicin Hepatotoxicity

As the primary medications for the treatment of Mycobacterium tuberculosis, hepatotoxicity occurs directly from isoniazid (INH) and indirectly from rifampicin (RMP) metabolites [244]. Evidence has been presented that the hepatic injury and subsequent fibrosis involve OS. The in vivo animal study demonstrated that activation of hepatic stellate cells and liver fibrosis depends on OS.

### 11.6. Amanitin Hepatotoxicity

Hepatotoxicity and death from mushroom poisoning mainly occur from the ingestion of *Amanita phalloides*. The toxic amanitins cause direct and indirect (through OS) hepatorenal injury. Currently, there is no definitive treatment for amanita poisoning [245]. Except for forced diuresis, care is essentially supportive. Perhaps future treatment studies will examine a combined approach—forced diuresis to hasten the renal clearance of amanitins, cholestyramine to block its entero-hepatic reuptake, and GSH to target OS.

## 12. Ocular Diseases

OS has been implicated in the pathogenesis of many ocular diseases. These retinopathies may be of degenerative, genetic, senile, or inflammatory origin. Among the resulting diseases are age-related macular degeneration (AMD), diabetic retinopathy (DM), retinitis pigmentosa (RP), retinitis of prematurity, retinal degeneration, and Stargardt disease [246].

### 12.1. Diabetic Retinopathy

Diabetic retinopathy (DR) is the most common microvascular complication in diabetic patients and is the primary cause of blindness in people between 27 and 75 years of age [247,248]. It results from retinal vascular abnormalities, including hyperpermeability, hypoperfusion, and neoangiogenesis. Kusuhara et al. have outlined the classification and management of DM [247]. Interestingly, a subpopulation of diabetic patients does not develop DR despite having the risk factors of sustained hyperglycemia, hypertension, hyperlipidemia, and pregnancy [247]. This suggests the presence of an additional unidentified risk factor. Finally, if OS initiates and promotes inflammation associated with DR, early antioxidant intervention may prevent its onset or slow its progression [248].

### 12.2. Retinitis Pigmentosum

Retinitis pigmentosum (RP) is an inherited, progressive retinal degeneration of the eyes. The involvement of OS in the complex pathologic progression of RP has been detailed [249]. This includes damage to cone photoreceptor morphology, the blood-retinal barrier, and upregulation of retinal expression of vascular endothelial growth factor (VEGF). In addition, a significant loss of GSH and associated high levels of MDA with the aqueous humor was noted. The high level of MDA reflects lipid peroxidation that promotes cytokine-associated inflammation and angiogenesis [250].

### 12.3. Age-Related Macular Degeneration

Age-related macular degeneration (AMD) is a progressive and irreversible ocular disease that is a significant cause of visual impairment in elderly individuals and is the fourth leading cause of blindness worldwide [251]. The Beckman clinical classification of AMD was reviewed. The retina’s high oxygen metabolism compared with any other tissue and its generation of ROS make it susceptible to OS. Animal studies have supported this premise. Ruan et al. made a detailed presentation of OS in the development of AMD [251]. Although the incidence has declined with the introduction of anti-VEGF agents, OS appears to be the initial trigger of AMD and influences late-stage neovascularization [252].

## 13. Systemic Inflammatory Response Syndrome and Sepsis

Systemic inflammatory response syndrome (SIRS) and its infection-associated variant known as sepsis involves a complex pathologic process that can progress to multi-organ failure (MOF) [253,254]. In addition to antibiotics, current sepsis management is supportive (mechanical ventilation, fluid administration, and pressors). Bar-Or et al. [253] hypothesized that tissue damage may be a consequence of IRI. If so, treating OS could be an additional treatment modality in SIRS/sepsis management. 

## 14. Pancreatitis

### 14.1. Acute Pancreatitis

Acute pancreatitis (AP) is usually caused by excessive alcohol consumption or bile stone occlusion of the pancreatic duct and, to a lesser extent, by extremely high triglyceridemia. The initial insult causes injury to pancreatic acinar cells, leading to increased ROS and OS production. This subsequently results in a dysregulated cytokine response and potentially SIRS [255]. The Atlanta Classification identifies two types of AP. There is an initial interstitial edematous AP and a severe necrotizing AP. Although beneficial, the current management of acute pancreatitis is supportive care. Antioxidant therapy has produced mixed results. However, studies with experimental models have demonstrated that OS is present from the early stage of the disease, and its degree (as measured by elevated MDA and c-reactive protein levels and inversely by low antioxidant levels) is associated with the severity of AP [256,257]. Interestingly, pretreatment with current antioxidants to prevent AP from endoscopic retrograde cholangiopancreatography (ERCP) has been unsuccessful. Given the uncertainty of the efficacy of numerous antioxidants (including NAC and vitamin C), the role of GSH remains to be determined.

### 14.2. Chronic Pancreatitis

In a comparison of 35 healthy subjects with 35 chronic pancreatitis (CP), multiple measures of both OS and antioxidant capacity were used [258]. Although the presence of OS was demonstrated, its significance in initiating and maintaining CP also remains to be determined.

## 15. Discussion

The evidence has been presented that supports the association of OS with numerous seemingly disparate illnesses. The association of OS with illnesses presents more questions than answers. The unanswered questions are the following: (1) Is OS only an associative biomarker? (2) Is OS a significant initiator and promoter? (3) Is OS present as a result of GSH deficiency? (4) Can GSH be restored and reverse OS? (5) As illustrated in Figure 4, can the early use of the G/C complex prevent the onset of progressive illnesses and avoid irreversible pathology? Due to the inverse relationship of OS with GSH levels, either due to age, metabolic status, or acute illness, a deficiency of GSH appears to result in OS. Given this premise, restoring GSH should reverse OS. Studies with numerous other antioxidants have been problematic to administer, equivocal, or disappointing [83,258]. However, most are weak antioxidants, and NAC is a GSH precursor that requires endogenous enzymatic conversion into GSH. Experimental antioxidants are not available for clinical use. Many immunotherapeutic agents are costly and are not readily administered. It is proposed that the transdermal G/C complex can deliver exogenous GSH effectively. The complex has many advantages over other antioxidants and treatments: (1) through nanotechnology, it delivers GSH directly in a measured amount; (2) it avoids the delivery problems of oral and IV GSH; (3) it avoids enzymatic conversion required by NAC; (4) as a topical agent, it is easily administered; (5) it has been commercially available for over ten years; (6) it does not require special storage; (7) relative to immunotherapeutic agents, it is a low-cost product. 

Given the complex pathophysiology of the many illnesses discussed in this review, the unanswered question is, what influence does OS have in the various disease processes? Is it a marker or a participant? If it is a participant, what effect will the G/C complex have in preventing or modifying OS? To answer these many questions, the complex needs to be rigorously studied to establish the efficacy of GSH in managing OS and to support or refute the role of OS pathophysiology in those illnesses. Significantly, in neurological illnesses, the CD can cross the blood–brain barrier. If future studies support the efficacy of the G/C complex, it could be used as an adjunct or alternative product to prevent or treat illnesses associated with OS that currently lack optimal intervention. Among the proposed studies are investigations into whether the complex can (1) treat the rare individuals with congenital absence of the enzymes GCL or GS, (2) modify the rapid aging process of H-G progeria or Werner’s syndrome, (3) prevent or reduce the consequences of IRI injury, (4) prevent the onset of progressive neurodegeneration, (5) attenuate the process of thromboinflammation in ARDS, cirrhosis, and CAD, (6, [196]) improve the management of diabetes and its complications, and (7) improve the treatment of Mycobacterium infections. Finally, can the complex contribute to the avoidance or management of severe acute insults from COVID-19, SIRS, pancreatitis, ischemia, toxicities, and virus-associated complications? 

## 16. Summary

There are numerous disparate acute and chronic illnesses associated with OS. Essentially, acute insults overwhelm cellular antioxidants, and chronic ones exhaust them, leading to excessive ROS, which results in OS. Although many illnesses have a documented association with OS and antioxidant deficiency, the efficacy of antioxidant replacement, including GSH, has been varied. Preliminary evidence has been presented of a compound, the G/C complex, that appears to deliver GSH effectively and therapeutically. Since the pathophysiology of illnesses is inherently complex and variable, presenting GSH administered via the G/C complex as the therapeutic “answer” is evocative but unrealistic. However, rigorous studies with the G/C complex may answer many questions about the role and efficacy of GSH in addressing OS and the intrinsic influence of OS in illnesses. It should be used in studies to determine if OS is solely an associated marker of illnesses or is an initiator and promoter of those illnesses. Hopefully, the complex will find its place as an adjunct or alternative to current therapeutic regimens in the management of untreatable or sub-optimally treated illnesses.

## Figures and Tables

**Figure 1 antioxidants-13-01106-f001:**
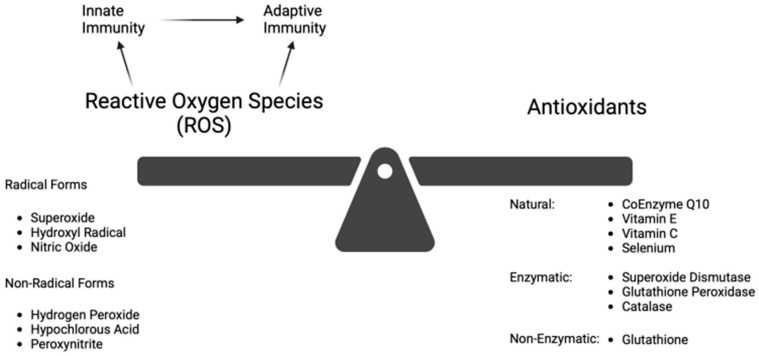
This illustrates the beneficial role of reactive oxygen species (ROS) when it is in physiologic balance with antioxidants. These include activating innate immunity and augmenting adaptive immunity. It should be noted that peroxynitrite and nitric oxide are co-existing reactive nitrogen species (RNS).

**Figure 2 antioxidants-13-01106-f002:**
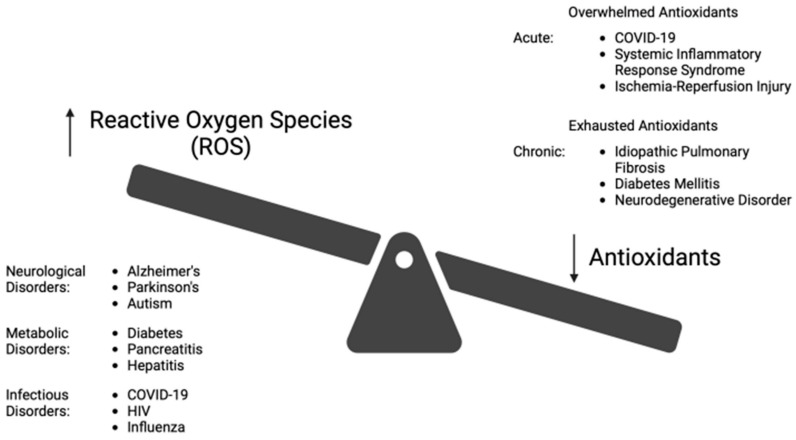
This illustrates the state of oxidative stress (OS) that occurs from an imbalance of ROS relative to antioxidants. It can result from excess ROS, a deficiency of antioxidants, or a combination of both. Acute insults occur when the stimulation of ROS overwhelms available antioxidants. Chronic insults occur when antioxidants are either deficient or exhausted. Presumptively, the development of numerous seemingly disparate acute and chronic diseases may be the result of OS. (Created with BioRender).

**Figure 3 antioxidants-13-01106-f003:**
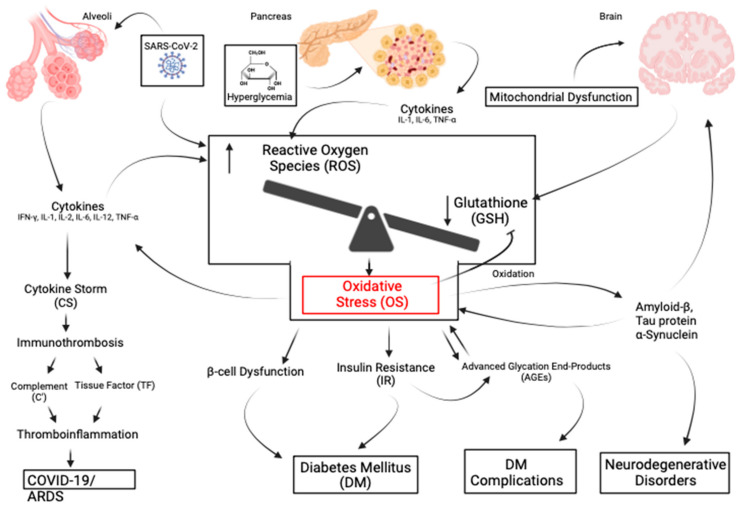
Illustrated are examples of acute and chronic diseases associated with OS. An acute disease resulting from OS is COVID-19. OS leads to a dysregulation of the cytokine response, which leads to the cytokine storm (CS). The CS causes multi-organ failure (MOF), including COVID-associated acute respiratory distress syndrome (ARDS). Chronically, OS causes beta-cell dysfunction and insulin resistance, leading to diabetes and its many neuro-vascular complications that result from the formation of advanced glycation end-products (AGEs). The onset of neurogenerative disorders appears to be initiated and enhanced by OS. (Created with BioRender).

**Figure 4 antioxidants-13-01106-f004:**
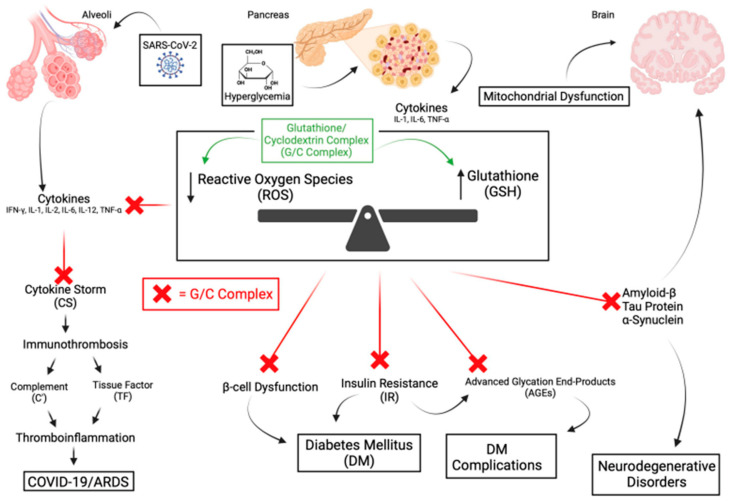
This illustrates the concept that the prevention or early inhibition of OS is a more effective approach to managing diseases. This should be achieved by targeting diseases early when the pathophysiologic process is more easily controlled and prior to the development of irreversible tissue injury. (Created with BioRender).

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
