# Peer review of "Treatment of Acute and Long-COVID, Diabetes, Myocardial Infarction, and Alzheimer’s Disease: The Potential Role of a Novel Nano-Compound—The Transdermal Glutathione–Cyclodextrin Complex"

_antioxidants, 2024, doi:10.3390/antiox13091106_

Round 1

Reviewer 1 Report

The paper of Yutani et al. focuses on Treatment of acute and long-COVID, diabetes, myocardial infarction, and Alzheimer’s disease. The potential role of a novel nano-compound - the transdermal glutathione-cyclodextrin complex.

The topic is interesting, but the paper need to be improved before acceptance.

-              The topic of the article is highly relevant given the current global health challenges, particularly with the ongoing impacts of COVID-19. The abstract is concise but it’ll be useful to include specific conclusions from the article to highlight the outcome of the study.

-              The introduction should be improved. You can briefly mention the specific benefits of related novel nano-compounds.

-              There are a couple of redundant lines in the “Discussion”. Please make it concise and expand more on future directions.

-              The language used is clear and precise, but the english can be improved – especially in chapter 2.

-              The general structure of the article is good but it is useful to note that not a lot of the references are recent. Especially for a novel topic like this, please prioritise references from postCOVID.

Author Response

We appreciate the thoughtful , constructive, and informative feedback of the reviewers.

  1. Hopefully, we have made the abstract clearer and more succinct.
  2. The introduction has been revised and hopefully provides more clarity
  3. The redundancy has been addressed
  4. Hopefully the revised language is satisfactory
  5. Nearly all references are from 2022 or later. We are not clear of the request to “prioritize” the references. FYI, the program, EndNote, was used to enter the bibliographical citations. We can send an EndNote copy upon request, if needed

Reviewer 2 Report

 Treatment of Acute and Long-COVID, Diabetes, Myocardial Infarction, and Alzheimer’s Disease. The Potential Role of a Novel Nano-Compound—The Transdermal Glutathione-Cyclodextrin Complex, reviews the role of glutathione in control of oxidative stresses encountered in several conditions, suggesting a superior method of delivery for GSH.

The discussion of oxidative stress leaves out a few items.  Peroxiredoxins/thioredoxins should at least be mentioned.  PRDX6 is a glutathione peroxidase having a catalytic cycle that involves GST-pi and GSH.

 Also, GSTs should be mentioned as major consumers of GSH promoting loss to export of GSH conjugated agents and having some role in control of electrophilic agents that might induce expression of antioxidants and GSH via NRF2.  Depending on the levels of ROOH, GPXs are generally less impacted by small to moderate reductions in GSH levels while GSTs would be affected.  However, at very high ROOH levels (>10-7 M) it has been suggested that GPXs could be limited at the reduction phase particularly in tissues where GSH levels are lower than the commonly quoted 1-5mM range; Ng CF, Schafer FQ, Buettner GR, Rodgers VG. The rate of cellular hydrogen peroxide removal shows dependency on GSH: mathematical insight into in vivo H2O2 and GPx concentrations. Free Radic Res. 2007 Nov;41(11):1201-11. doi: 10.1080/10715760701625075. PMID: 17886026; PMCID: PMC2268624.

The role of glutathionylation of proteins as a protective effect of GSH in OS could be brought up.

Since you discuss diabetes millitis and GSH, you could mention succination as a pathology that would affect GSH levels (one target along with free sulfhydryls in proteins, including thioredoxin) Frizzell N, Lima M, Baynes JW.  Succination of proteins in diabetes. Free Radic Res. 2011 Jan;45(1):101-9. doi: 10.3109/10715762.2010.524643. Epub 2010 Oct 22. PMID: 20964553.

 Sources of OS also include NADPH oxidases in the cell membrane and possibly the ER and phagosomes. 

With these items included, the review will have more impact.

See above

Author Response

Replies to reviewers.

We appreciate the thoughtful , constructive, and informative feedback of the reviewers.

  1. The significance of peroxidase and peroxiredoxins has been recognized and included in more detail. Since this manuscript is focused from a clinical perspective, their clinical significance in diseases is emphasized. It adds to the robustness of this review and presented a biochemical explanation of many diseases processes. Hopefully, it satisfies the reviewer. Thanks for the recommendations.
  2. Likewise, the influence of GSTs was added.
  3. The process of glutathionylation was added.
  4. Thanks for the recommendation to address the role of succinate and succination in diseases and disease risks. I agree with with the author, S Fernandez-Veledo, that it “connects the dots”
  5. The contribution of the endoplasmic reticulum was briefly mentioned in the second paragraph of “ROS and OS” with the reference by Di Meo. We recognize the significance of NAPD oxidases but did not delve into the biochemical details of OS since the focus of the manuscript was disease management from a clinician’s perspective.

Round 2

Reviewer 1 Report

The authors well replied to my previous comments, nothing to add.

The authors well replied to my previous comments, nothing to add.

Author Response

Once again, we appreciate your thoughtful, constructive feedback. 

Reviewer 2 Report

With the additions, in general, the paper adequately reviews the topic and provides important information, with one minor point.

The paragraph on succination needs a little revision. The actual offending metabolite is fumarate, not succinate.  After the attack on the sulfhydryl group to form a thioether linkage, it is referred to as succinate.  Also, explain that GSH is one target of the process.

Author Response

Once again, we appreciate your thoughtful, constructive, and informative feedback. Per your recommendation we made those edits highlighted in yellow. Look forward to publishing this work soon. Thank you.